# Effect of Light Irradiation on the Diffusion Rate of the Charge Carrier Hopping Mechanism in P3HT–ZnO Nanoparticles Studied by μ+SR

**Eka Pratikna [1], Lusi Safriani [1,\*], Nowo Riveli [1], Budi Adiperdana [1], Suci Winarsih [1], Annisa Aprilia [1], Dita Puspita Sari [2], Isao Watanabe [3] and Risdiana Risdiana [1]**

[1] Department of Physics, Faculty of Mathematics and Natural Sciences, Padjadjaran University, Jl. Raya Bandung-Sumedang Km. 21, Jatinangor, Sumedang 45363, Indonesia; eka19010@mail.unpad.ac.id (E.P.); nowo@phys.unpad.ac.id (N.R.); budi.adiperdana@phys.unpad.ac.id (B.A.); suci.winarsih@unpad.ac.id (S.W.); a.aprilia@phys.unpad.ac.id (A.A.); risdiana@phys.unpad.ac.id (R.R.)

[2] Graduate School of Engineering and Science, Shibaura Institute of Technology, 307 Fukasaku, Minuma, Saitama 337-8570, Japan; dita.puspita.sari.t1@shibaura-it.ac.jp

[3] Meson Science Laboratory, RIKEN Nishina Center, 2-1 Hirosawa, Wako, Saitama 351-0198, Japan; nabedon@riken.jp

\* Correspondence: lusi.safriani@phys.unpad.ac.id

**Abstract:** Blended regio-regular P3HT–ZnO nanoparticles are a hybrid material developed as an active layer for hybrid solar cells. The study of the hopping mechanisms and diffusion rates of regio-regular P3HT–ZnO nanoparticles is significant for obtaining intrinsic charge transport properties that provide helpful information for preparing high-performance solar cells. The temperature dependences of the parallel and perpendicular diffusion rates in regio-regular P3HT–ZnO nanoparticles determined from muon spin relaxation measurements were investigated by applying various longitudinal fields. We investigated the effect of light irradiation on the diffusion rates in regio-regular P3HT–ZnO nanoparticles. We found that with increasing temperature, the parallel diffusion rate decreased, while the perpendicular diffusion rate increased. The ratio of the parallel to perpendicular diffusion rate ($D_\parallel/D_\perp$) can be used to indicate the dominant charge carrier hopping mechanism. Without light irradiation, perpendicular diffusion dominates the charge carrier hopping, starting at 25 K, with a ratio of $1.70 \times 10^4$, whereas with light irradiation, the perpendicular diffusion of the charge carrier starts to dominate at the temperature of 10 K, with a ratio of $2.40 \times 10^4$. It is indicated that the additional energy from light irradiation affects the diffusion, especially the charge diffusion in the perpendicular direction.

**Keywords:** diffusion rate; hybrid solar cells; LF-μ+SR; light irradiation; P3HT–ZnO nanoparticles

## 1. Introduction

Photovoltaic solar cells are very promising devices for development as a tool for converting solar energy into electrical energy. Various studies have been carried out to optimize the utilization of solar energy, either by modifying the material in photovoltaic solar cells [1–6] or by designing optical devices that can increase the maximum utilization of solar energy harvested at any time in various natural conditions [7].

The materials developed as photovoltaic materials consist of organic, inorganic, and hybrid materials. Each of these materials has its own unique characteristics and its own advantages and disadvantages. Organic solar cells exhibit great advantages, such as lightweightedness, allowing easy fabrication, and low cost. However, their power conversion efficiency (PCE) is still lower than that of inorganic solar cells. Recently, PCE as high as 18.4% was reported for organic solar cells [8], which is still lower than that for inorganic silicon heterojunction solar cells, with a PCE of 26% [3]. There have been many attempts to improve the PCE of organic solar cells, such as modifying the active material of solar

cells [4]. Among organic materials, conjugated polymer materials have been widely used as active materials in solar cell devices. In particular, during the last few years, poly(3-hexylthiophene) (P3HT) has been used as a polymer donor in solar cell research [5]. P3HT is widely used because of its light absorption as well as higher hole mobility compared with other conjugated polymers [6,9]. In addition to PCE, organic-based solar cell devices still have some deficits in their performance, such as weak absorption at visible wavelengths, poor charge transport, and low stability [5]. To overcome the various deficits of organic solar cells and to improve their performance, many researchers have developed solar cells with materials derived from combinations of organic and inorganic materials known as hybrid solar cells [10].

The development of hybrid solar cells is attracting great interest because of the various advantages of this material [11]. Hybrid solar cells combine the superior properties of organic and inorganic materials. Organic materials, such as P3HT, are used as donors to absorb sunlight and transport holes, while inorganic materials, such as semiconductors of CdSe, $TiO_2$, and ZnO, are used as acceptors to transport electrons [10]. Previous studies have revealed that inorganic materials of nanoparticle size enhance optical absorption [12]. One material currently widely used is ZnO because it is easy to synthesize ZnO at nanoparticle sizes [13,14]. Another advantage of ZnO is that it is nontoxic compared with CdSe [10]. For electron mobility, ZnO has a higher value ($2 \times 10^{-3}$ cm$^2$/V·s) than $TiO_2$ ($1 \times 10^{-4}$ cm$^2$/V·s) [15]. A high electron mobility facilitates efficient electron transport and produces high-efficiency solar cells [15–17]. In addition, ZnO nanoparticles have a lower conduction band than the lowest unoccupied molecular orbital (LUMO) of P3HT, which facilitates the transport of electrons from donor to acceptor [18]. Thus, a combination of P3HT and ZnO nanoparticles can be used to achieve the expected and improved performance in hybrid solar cells [16,17], although the PCE for the hybrid solar cells is still less than 1% [19].

One important characteristic of hybrid solar cells that needs to be investigated is the charge carrier mobility. The charge carrier mobility is a crucial parameter for optimizing solar cell performance, especially related to charge extraction and charge carrier recombination [20]. The value of the charge carrier mobility can be obtained from electron spin resonance (ESR) or nuclear magnetic resonance (NMR) measurements. However, these two measurements require a high concentration of doping in order to obtain the mobility value; this renders the intrinsic properties of the undoped material unable to be observed properly. To obtain the value of the charge carrier mobility in a material with a low doping concentration, muon spin relaxation ($\mu^+$SR) measurement is a suitable technique [21,22]. When a muon is implanted into the polymer sample, it interferes with the carbon double bonds in the polymer and forms muonium (the bound state of the positively charged muon and electrons), which gives rise to unpaired electrons and an accompanying polarization field, called the negative polaron, in the polymer chain. When the longitudinal magnetic field is applied to the sample, the polaron moves away and diffuses parallel (intra) and perpendicular (inter) to the polymer chain [23–26].

In a previous $\mu^+$SR study, an active layer of hybrid solar cells of regio-regular P3HT–ZnO nanoparticles was measured with light irradiation, and it was found that at 10 K, the charge carrier transport was dominated by three-dimensional interchain diffusion [27,28], which differs from the $\mu^+$SR measurements without light irradiation, in which the three-dimensional diffusion was observed at the higher temperature of 25 K [29]. However, the values of the intrachain and interchain diffusion rates for the blended regio-regular P3HT–ZnO nanoparticles have not been reported. The diffusion rate of charge carrier transport needs to be estimated to obtain information about the number of charge carriers that can be transported either intra or interchain in order to estimate the solar cell's efficiency. This is because solar cells' efficiency strongly depends on the charge carrier mobility, which correlates with the diffusion rate. Hence, it is necessary to determine the diffusion rate in order to obtain solar cells with high efficiency.

In this paper, we report the diffusion rates of blended regio-regular P3HT–ZnO nanoparticles determined using longitudinal field muon spin relaxation (LF-μ$^+$SR), and the effect of light irradiation on the diffusion rate. By analyzing the μ$^+$SR spectra, the hopping mechanism and the diffusion rate of the charge carriers in the sample can be calculated. The calculation and analysis were based on an empirical function using the equations relating to the relaxation rate data as a function of the $H_{LF}$ with the diffusion rates [30–33]. This paper is organized as follows: Section 1 is an introduction describing the state of the diffusion rates and the reason for choosing the regio-regular P3HT–ZnO nanoparticles as the material to be investigated. Section 2 explains the method for the material preparation and measurement of the μ$^+$SR as well as the calculations and data analysis based on empirical functions. Section 3 presents the main results from the μ$^+$SR data and discusses the analysis of the diffusion rate calculations. Finally, we present conclusions from this research in Section 4.

## 2. Materials and Methods

High-quality regio-regular P3HT was obtained from Sigma-Aldrich and used without further purification. ZnO nanoparticles were synthesized by the sol-gel method using zinc acetate dehydrate (Merck, >99.9%) as a precursor, with sodium hydroxide (Merck, >99.9%) as a catalyst, and in methanol, as a solvent [34]. The relative amount of ZnO nanoparticles was 20 wt% for the regio-regular P3HT. The process of mixing regio-regular P3HT and ZnO nanoparticles was as follows: regio-regular P3HT (150 mg) was dissolved in chlorobenzene (2.91 mL) and stirred at room temperature to obtain homogeneous solutions. Separately, ZnO nanoparticles (30 mg) were dispersed in methanol (0.09 mL) followed by dispersion in an ultrasonic bath for 10 min. The ZnO nanoparticles dispersed in solution were added to the regio-regular P3HT solution, and the mixture was stirred for 2 h at 40 °C. Bulk samples of regio-regular P3HT–ZnO nanoparticles were obtained by evaporating the solutions at 150 °C for 2 h [14].

We performed μ$^+$SR experiments in various longitudinal fields (LF) ranging from 0 to 395 mT, and at temperatures of 10 K, 15 K, 25 K, 50 K, and 300 K for samples of regio-regular P3HT–ZnO nanoparticles in conditions with and without white light irradiation (with a pulse-type flash lamp, 130 W, with 50 J per flash) at the RIKEN-RAL Muon Facility in the UK [27,28]. We collected 30 million events for every dataset.

The data obtained from μ$^+$SR measurements were the time-dependent asymmetry, and they were fitted using a simple exponential function as shown in Equation (1) [26–29,35–39].

$$A\,(t)\ =\ A_1 \exp(\lambda_1 t)\ +\ A_2 \exp(\lambda_2 t) \tag{1}$$

The asymmetry function is divided into the fast and slow components. $A_1$ and $A_2$ are the initial asymmetries; $\lambda_1$ and $\lambda_2$ are the depolarization rates of the fast and slow components, respectively.

From Equation (1), the values of $\lambda$ for each field ($H_{LF}$) at a certain temperature were obtained. The dependence of $\lambda$ on $H_{LF}$ can be refitted using the function of Equation (2) or (3) to obtain information on the direction of charge carrier hopping as previously reported [26–31,35–39]. If the field dependence ($H_{LF}$) of $\lambda$ can be fitted properly using Equation (2), then the charge carrier hopping can be defined as one-dimensional intrachain hopping diffusion.

$$\lambda = \mathrm{c} H_{LF}{}^{-0.5} \tag{2}$$

If the field dependence ($H_{LF}$) of $\lambda$ can be fitted properly using Equation (3), the charge carrier hopping can be defined as three-dimensional interchain hopping diffusion.

$$\lambda = \mathrm{c} - H_{LF}{}^{0.5} \tag{3}$$

The diffusion rate calculations were extracted from the equation for the relaxation rate as a function of the $H_{LF}$, as shown in Equation (4) [30].

$$\lambda(H_{LF}) = \rho(\Omega^z f(\omega_\mu) + \Omega^+ f(\omega_e)) \tag{4}$$

where $\omega = \gamma H$, $\gamma$ are gyromagnetic ratios, $\rho$ is the spin density, $\Omega^z$ and $\Omega^+$ are geometrical factors that are related to the scalar and dipolar hyperfine coupling parameters, and $f(\omega)$ is the spectral density of the spin correlation function as previously reported for calculating the diffusion rate in polymer samples, as shown in Equation (5) [31–33].

$$f(\omega) = \frac{1}{\sqrt{4D_\parallel D_\perp}} \sqrt{\frac{1 + \sqrt{1 + \left(\frac{\omega}{2D_\perp}\right)^2}}{1 + \left(\frac{\omega}{2D_\perp}\right)^2}} \tag{5}$$

The values of the parameters used in the calculation are $\rho\Omega^z = (150\text{ MHz})^2$, $\Omega^+/\Omega^z = 14$, $\gamma_\mu = 135.5387\text{ MHz/T}$, and $\gamma_e = 28{,}024.95164\text{ MHz/T}$ [30–33].

For the parallel diffusion rate ($D_\parallel$), the value of $D_\parallel$ can be obtained by examining the function in Equation (6). The high $H_{LF}$ dependence of $\lambda$ as shown in Figure 1 can be well fitted using Equation (2).

$$D_\parallel = \frac{2\rho^2}{c^2}\left(\frac{\Omega^z}{\sqrt{\gamma_\mu}} + \frac{\Omega^+}{\sqrt{\gamma_e}}\right)^2 \tag{6}$$

where $c$ is the gradient in Equation (2) that was acquired from the fitting method.

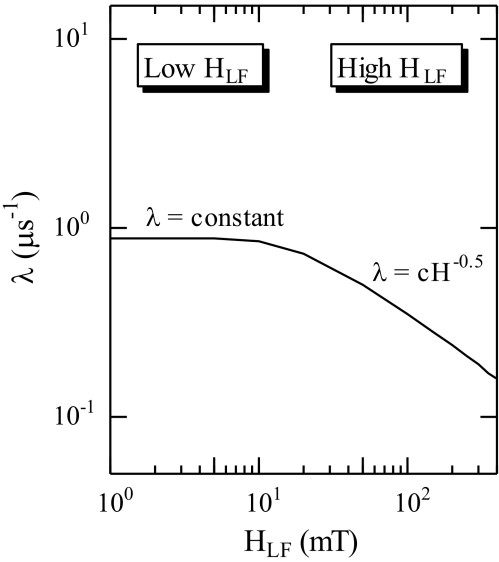

**Figure 1.** Two regions of longitudinal field $H_{LF}$ dependence of relaxation rate $\lambda$ for high $H_{LF}$ and low $H_{LF}$.

For the perpendicular diffusion rate ($D_\perp$), the value of $D_\perp$ has to be determined using Equation (7). At low $H_{LF}$, the $\lambda$ value is approximately constant as shown in Figure 1.

$$D_\perp = \frac{2\rho^2(\Omega^z + \Omega^+)^2}{\lambda_0 D_\parallel} \tag{7}$$

where $D_\parallel$ is already obtained following the previous explanation, and $\lambda_0$ is the average value of $\lambda$ in the constant region, or alternatively, $\lambda_0$ can be approximated as the value of $\lambda$ when $H_{LF}$ is zero.

## 3. Results and Discussion

Figure 2 shows time dependence of the asymmetry data for regio-regular P3HT–ZnO nanoparticles for various applied longitudinal fields with and without light irradiation at temperatures of 10 K, 15 K, 25 K, 50 K, and 300 K. All the asymmetry data were well fitted using Equation (1). The initial asymmetry shifted to higher values with increasing $H_{LF}$ as a result of the depolarization of the muonium state [40].

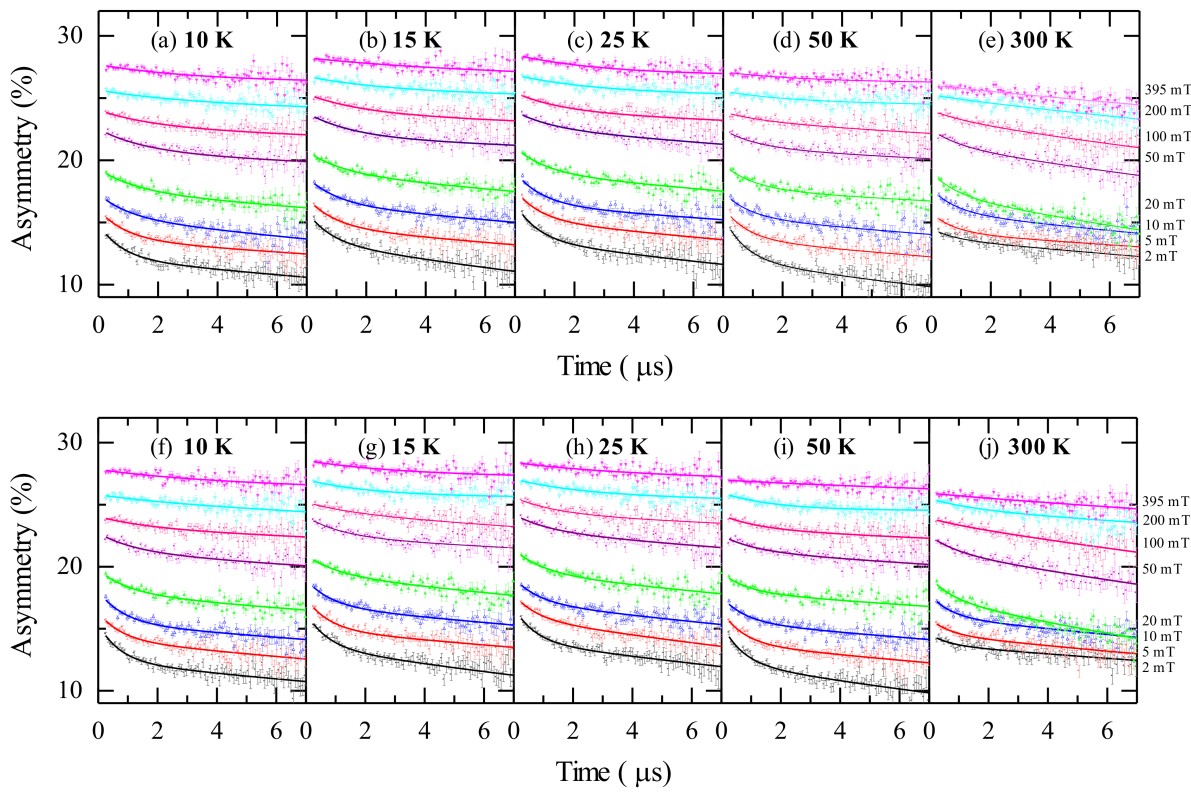

**Figure 2.** The asymmetry data for regio-regular P3HT–ZnO nanoparticles for various applied longitudinal fields without light irradiation at temperatures of (**a**) 10 K, (**b**) 15 K, (**c**) 25 K, (**d**) 50 K, and (**e**) 300 K, and with light irradiation at temperatures of (**f**) 10 K, (**g**) 15 K, (**h**) 25 K, (**i**) 50 K, and (**j**) 300 K.

To demonstrate the effect of light irradiation, asymmetry data with and without light irradiation at 15 K and 25 K for a $H_{LF}$ of 10 mT are shown in Figure 3. When using light irradiation, the asymmetry shifts slightly higher due to the addition of energy that affects the depolarization of the muonium state, which may also affect the transport of charge carriers.

Figure 4 shows the $H_{LF}$ dependence of the depolarization rate $\lambda_1$ of the regio-regular P3HT–ZnO nanoparticles with and without light irradiation at temperatures of 10 K, 15 K, 25 K, 50 K, and 300 K. The value of $\lambda_1$ was extracted from the fitting result for the asymmetry data using Equation (1). Without light irradiation, the dependence of $H_{LF}$ on $\lambda_1$ for temperatures of 10 K and 15 K showed a linear trend in a log–log plot that could be fitted using Equation (2), which indicates the direction of the charge carrier hopping of one-dimensional intrachain diffusion. For temperatures of 25 K, 50 K, and 300 K, the dependence of $H_{LF}$ on $\lambda_1$ is in accordance with Equation (3), which indicates the direction of the charge carrier hopping of three-dimensional interchain diffusion. With light irradiation, the dependence of $H_{LF}$ on $\lambda_1$ for all temperatures can be fitted using Equation (3), which shows the direction of the charge carrier hopping of three-dimensional interchain diffusion. It was found that, at temperatures of 10 K and 15 K, without light irradiation, the charge carrier hopping mechanism was dominated by one-dimensional intrachain diffusion. However, with light irradiation, the charge carrier hopping mechanism completely changed to three-

dimensional interchain diffusion, confirming that the additional energy of light irradiation affects the direction of charge carrier hopping. A similar tendency was observed for the $H_{LF}$ dependence of $\lambda_2$ with and without light irradiation at temperatures of 10 K, 15 K, 25 K, 50 K, and 300 K, as shown in Figure 5. Although the tendency was the same, the value of $\lambda_2$ was much smaller than $\lambda_1$, therefore, for the present study, the fast component in the short time region of the depolarization rate ($\lambda_1$) was analyzed in more detail, due to $\lambda_1$ providing more significant information than $\lambda_2$.

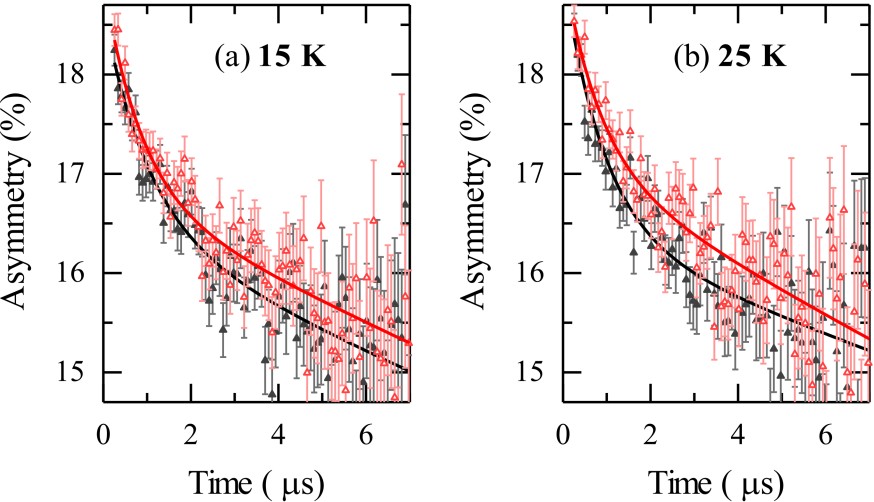

**Figure 3.** Different asymmetry data for regio-regular P3HT–ZnO nanoparticles for 10 mT with (△) and without (▲) light irradiation at temperatures of (**a**) 15 K and (**b**) 25 K.

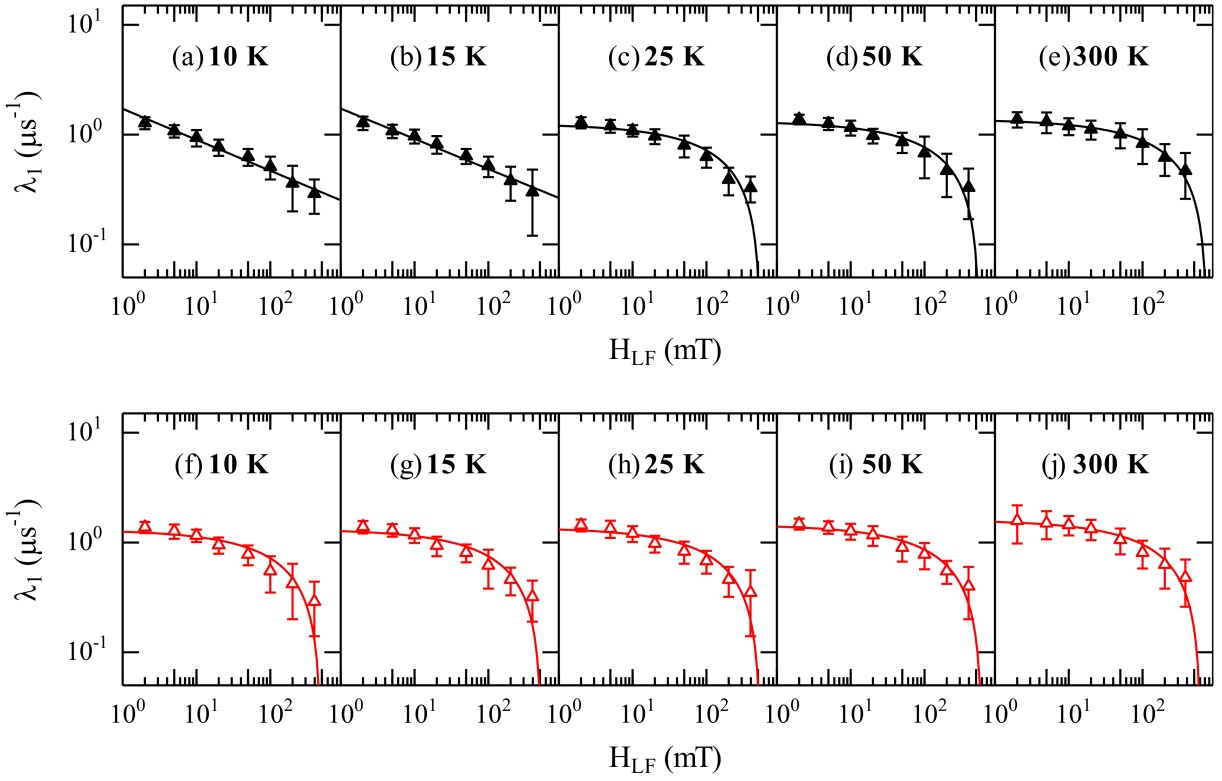

**Figure 4.** The longitudinal field dependence $H_{LF}$ of the relaxation rate $\lambda_1$ of the regio-regular P3HT–ZnO nanoparticles without light irradiation at temperatures of (**a**) 10 K, (**b**) 15 K, (**c**) 25 K, (**d**) 50 K, and (**e**) 300 K and with light irradiation at temperatures of (**f**) 10 K, (**g**) 15 K, (**h**) 25 K, (**i**) 50 K, and (**j**) 300 K.

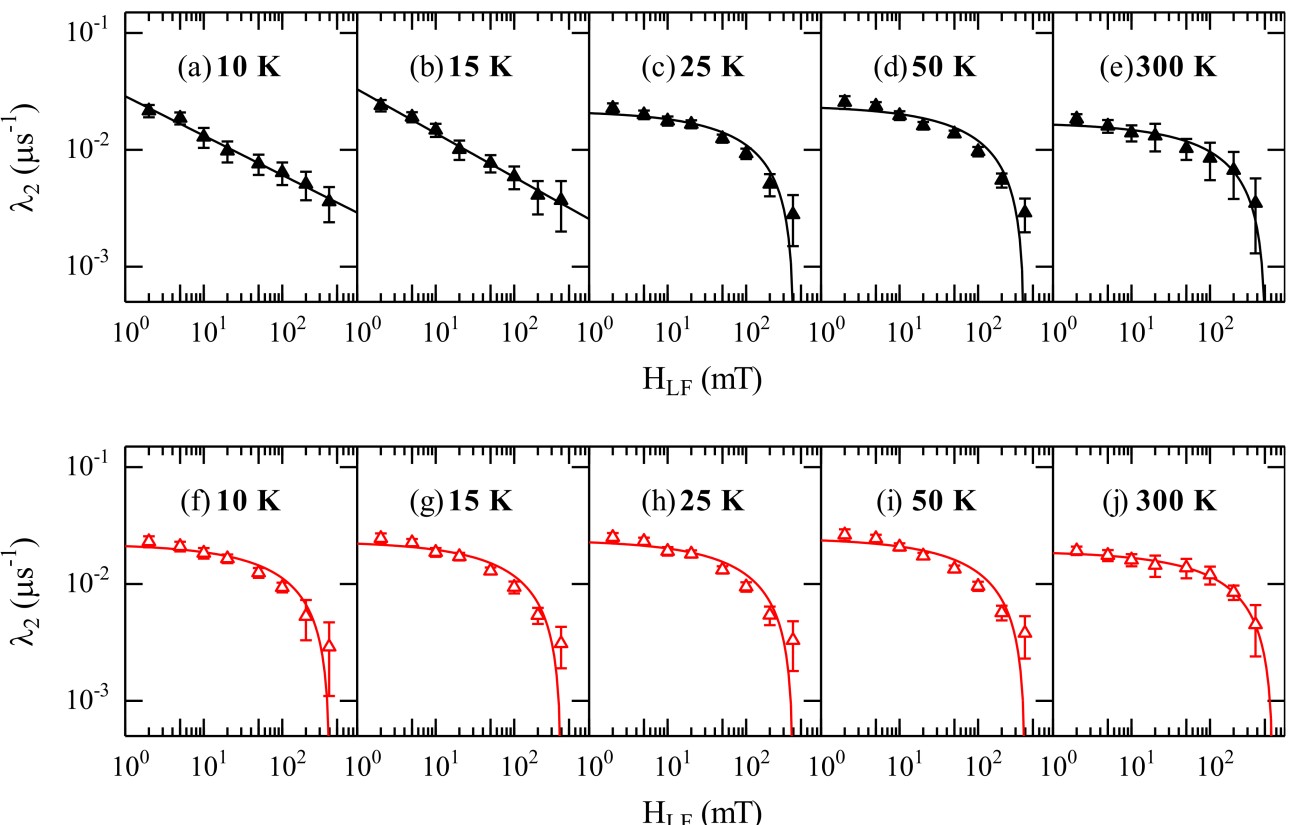

**Figure 5.** The longitudinal field dependence $H_{LF}$ of the relaxation rate $\lambda_2$ of the regio-regular P3HT–ZnO nanoparticles without light irradiation at temperatures of (**a**) 10 K, (**b**) 15 K, (**c**) 25 K, (**d**) 50 K, and (**e**) 300 K and with light irradiation at temperatures of (**f**) 10 K, (**g**) 15 K, (**h**) 25 K, (**i**) 50 K, and (**j**) 300 K.

After obtaining the information on the charge carrier hopping mechanism, the value of the diffusion rate can be calculated using Equations (6) and (7), as described in the methods section. The diffusion rates of the charge carrier in regio-regular P3HT–ZnO nanoparticles at temperatures of 10 K, 15 K, 25 K, 50 K, and 300 K with and without light irradiation are presented in Tables 1 and 2, respectively.

**Table 1.** The diffusion rates for regio-regular P3HT–ZnO nanoparticles at several temperatures without light irradiation obtained in this study.

|  | 10 K | 15 K | 25 K | 50 K | 300 K |
|---|---|---|---|---|---|
| $D_{\parallel}$ ($\times 10^{15}$ rad/s) | 1.39 | 1.35 | 0.19 | 0.16 | 0.09 |
| $D_{\perp}$ ($\times 10^{10}$ rad/s) | – | – | 1.15 | 1.25 | 1.73 |
| $D_{\parallel}/D_{\perp}$ ($\times 10^{4}$) | – | – | 1.70 | 1.28 | 0.57 |

**Table 2.** The diffusion rates for regio-regular P3HT–ZnO nanoparticles at several temperatures with light irradiation obtained in this study.

|  | 10 K | 15 K | 25 K | 50 K | 300 K |
|---|---|---|---|---|---|
| $D_{\parallel}$ ($\times 10^{14}$ rad/s) | 2.28 | 1.83 | 1.61 | 1.19 | 0.88 |
| $D_{\perp}$ ($\times 10^{10}$ rad/s) | 0.95 | 1.13 | 1.20 | 1.38 | 1.79 |
| $D_{\parallel}/D_{\perp}$ ($\times 10^{4}$) | 2.40 | 1.62 | 1.34 | 0.86 | 0.49 |

In the case of $\mu^{+}$SR measurement without light irradiation, the parallel diffusion rate ranged from $1.39 \times 10^{15}$ to $0.09 \times 10^{15}$ rad/s. The parallel diffusion rate decreased with increasing temperature. The perpendicular diffusion rates were obtained starting from

25 K. It was found that the values increased with increasing temperatures. These results are consistent with our proposed model regarding the change in dimensionality obtained from the $H_{LF}$ dependence of $\lambda_1$ as described in Figure 4. When comparing the diffusion rate with that for regio-regular P3HT without ZnO nanoparticles, we found that the parallel diffusion rate in the present study was of the same order of magnitude, while the perpendicular diffusion rate was two orders of magnitude higher in the high-temperature range [32]. Moreover, compared with those of other conjugated polymers such as polyaniline (PANI), polypyridine, and polyphenylenevinylene (PPV), the parallel diffusion rates of regio-regular P3HT–ZnO nanoparticles are higher by up to three orders of magnitude, and the perpendicular diffusion rates are slightly higher than those of PANI and PPV at high temperature [41,42]. The increase in the parallel and perpendicular diffusion rates indicates that the addition of ZnO nanoparticles facilitates the transfer of charge carriers due to the different bandgaps of P3HT and ZnO nanoparticles. The energy level (LUMO) of P3HT is −3.3 eV, while the conducting band of ZnO nanoparticles is −4.2 eV [28]. The low value of the conduction band of ZnO nanoparticles makes the diffusion process easier, resulting in high values for both parallel and perpendicular diffusion.

For μ+SR measurement with light irradiation, the values of both parallel and perpendicular diffusion were obtained for all temperatures starting from 10 K. At 10 K and 15 K, the perpendicular diffusion rates were $0.95 \times 10^{10}$ and $1.13 \times 10^{10}$ rad/s, respectively. These values were not observed in μ+SR measurement without light irradiation. The parallel diffusion decreased with increasing temperature, whereas the perpendicular diffusion rate increased with increasing temperature. The parallel diffusion rates ranged from $2.28 \times 10^{14}$ to $0.88 \times 10^{14}$ rad/s while the perpendicular diffusion rates ranged from $0.95 \times 10^{10}$ to $1.79 \times 10^{10}$ rad/s. When light irradiation was performed, the parallel diffusion rate was lower by one order of magnitude than the diffusion rate without light irradiation at 10 K and 15 K. Meanwhile, at 25 K to 300 K, the diffusion rate was of the same order of $10^{14}$, as shown in Figure 6. For μ+SR measurement with light irradiation, the perpendicular diffusion rate increased slightly with increasing temperature compared with that for μ+SR measurement without light irradiation. It is indicated that the additional energy from light irradiation affects the diffusion, especially the charge diffusion in the perpendicular direction.

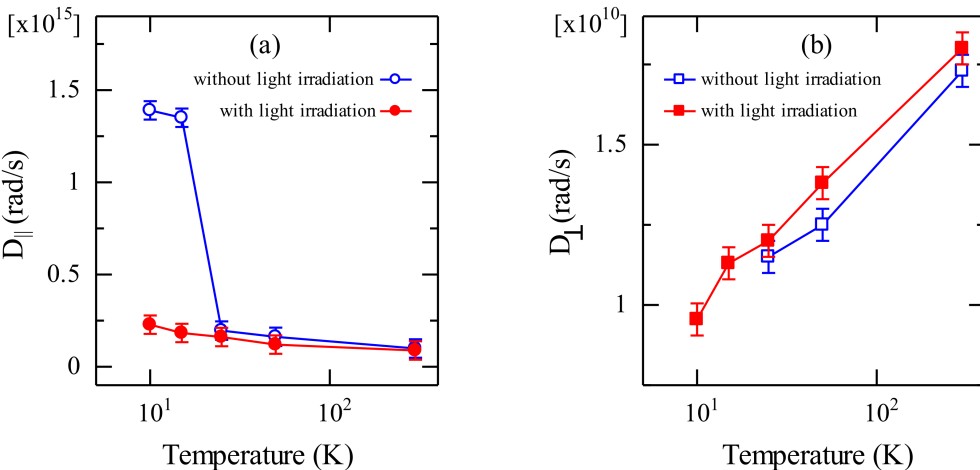

**Figure 6.** Temperature dependence of the diffusion rates in regio-regular P3HT–ZnO nanoparticles: (**a**) parallel and (**b**) perpendicular.

To elucidate the dominance of the charge carrier hopping mechanism, the ratio of the parallel and perpendicular diffusion rates ($D_{\parallel}/D_{\perp}$) was analyzed. Without light irradiation, the values of $D_{\parallel}/D_{\perp}$ at 25 K, 50 K, and 300 K were $1.70 \times 10^4$, $1.28 \times 10^4$, and $0.57 \times 10^4$, respectively. With light irradiation, the values of $D_{\parallel}/D_{\perp}$ at 10 K, 15 K,

25 K, 50 K, and 300 K were $2.40 \times 10^4$, $1.62 \times 10^4$, $1.34 \times 10^4$, $0.86 \times 10^4$, and $0.49 \times 10^4$, respectively, as shown in Tables 1 and 2. It was found that the value of $D_\parallel/D_\perp$ had an order of $10^4$, corresponding to the domination of charge carrier interchain diffusion. The present result is also consistent with the results reported for the PANI:EB material, where the value of $D_\parallel/D_\perp$ was $2.5 \times 10^4$ at a temperature of 150 K for the interchain diffusion of polaron motion [43]. The analysis of the $D_\perp$ in this study showed that the diffusion rate increased with increasing temperature. The tendency of $D_\perp$ at temperatures above 300 K will follow an increasing trend as does $D_\perp$ at temperatures of 10 to 300 K. Thus, the model proposed in this study is believed to be suitable for calculating the diffusion rate at temperatures above 300 K. The increase in $D_\perp$ above 300 K correlated to an increase in solar cell efficiency, which will provide advantages for application in tropical countries where the average temperature is above 300 K.

## 4. Conclusions

The hopping mechanism and diffusion rate of regio-regular P3HT–ZnO nanoparticles were studied to obtain $D_\parallel$ and $D_\perp$ by $\mu^+$SR measurements with and without light irradiation. Without light irradiation, $D_\parallel$ decreased with increasing temperature, while $D_\perp$ was observed starting from 25 K and increased with increasing temperature. On the other hand, with light irradiation, the $D_\perp$ observed from 10 K indicated that the additional energy from light irradiation affected the diffusion, especially the charge diffusion in the perpendicular direction. The value of $D_\parallel/D_\perp$ for $\mu^+$SR measurement with and without light irradiation ranged from $2.40 \times 10^4$ to $0.49 \times 10^4$ and $1.70 \times 10^4$ to $0.57 \times 10^4$, respectively. A smaller value of $D_\parallel/D_\perp$ indicates a higher domination of charge carrier interchain diffusion.

The diffusion calculation model reported in this paper is a model based on the fitting method with two exponentials. This approach is very powerful for obtaining the diffusion rate in the parallel and perpendicular directions. The diffusion rate can be confirmed by measuring the conductivity as a function of temperature. This model has many advantages and good future prospects. The main advantage of this model is that two parameters, namely the dimensionality and diffusion rate, are obtained simultaneously. To date, the results of $\mu^+$SR measurements for active solar cell materials, such as P3HT, P3HT/PCBM, and P3HT–ZnO nanoparticles, have only determined the dimensionality of the charge carrier mobility; therefore, the diffusion rate has not been obtained. The value of the diffusion rate is required when a material is to be used in applications, especially for solar cell devices. By using this model, not only the dimensionality, but also the diffusion rate, can be exactly calculated and obtained from $H_{LF}$-$\mu^+$SR measurements. Thus, researchers can use these two parameters in designing materials for applications, especially materials for producing high-efficiency solar cells. In addition, by using this model, we can also determine the diffusion rates for other materials such as poly(3-alkylthiophene) derivatives (poly(3-butylthiophene) and poly(3-octylthiophene)), which also have the potential to be used as active materials in solar cells. If the diffusion rates of other materials calculated using this model are higher than those of P3HT–ZnO nanoparticles, then these materials will become strong candidates for the active materials of solar cells.

**Author Contributions:** Conceptualization, L.S., I.W. and R.R.; methodology, L.S. and N.R.; software, B.A.; validation, L.S., N.R. and R.R.; formal analysis, E.P., N.R., B.A. and S.W.; investigation, E.P., A.A., D.P.S. and I.W.; resources, A.A. and I.W.; data curation, L.S., D.P.S. and R.R.; writing—original draft preparation, E.P.; writing—review and editing, L.S., S.W., D.P.S. and R.R.; visualization, E.P.; supervision, L.S. and R.R.; project administration, I.W.; funding acquisition, R.R. All authors have read and agreed to the published version of the manuscript.

**Funding:** We would like to thank Kemenristek DIKTI for providing financial support with the World Class Research Grant No. 1207/UN6.3.1/PT.00/2021 and Padjadjaran University for providing support with the Academic Leadership Grant No. 1959/UN6.3.1/PT.00/2021.

**Institutional Review Board Statement:** Not applicable.

**Informed Consent Statement:** Not applicable.

**Data Availability Statement:** Not applicable.

**Conflicts of Interest:** The authors declare no conflict of interest.

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
