# Peer review of "Effect of Light Irradiation on the Diffusion Rate of the Charge Carrier Hopping Mechanism in P3HT–ZnO Nanoparticles Studied by μ+SR"

_energies, doi:10.3390/en14206730_

Round 1

Reviewer 1 Report

The manuscript is about the effect of light irradiation, the changes of (diffusion rate) D|| and D, as well as D||/D were investigated by using the measurement method µ+SR of solar cell. The manuscript was very well written in English, the data was fully and carefully presented in the manuscript.

Therefore, I recommend the manuscript to be accepted by the Energies journal.

Author Response

We would like to thank for some comments and suggestions from reviewers to our manuscript. 

Reviewer 2 Report

The research article entitled “Effect of Light Irradiation on the Diffusion Rate of the Charge Carrier Hopping Mechanism in P3HT: ZnO Nanoparticles Studied by μ+SR” is submitted by Pratikna et al. in the Energies. The authors investigated the effect of light irradiation on the diffusion rates in regio-regular P3HT: ZnO nanoparticles. Overall, the paper is well-written and it includes good results. However, I have some reservations and suggestions that need to be addressed.  

  • Please highlight the importance of choosing ZnO over TiO2, ZrO2, CdSe, etc. It should be emphasized in the introduction. Is the lower conduction band of ZnO than the LUMO of P3HT was the only reason for choosing the ZnO?
  • Please provide the complete experimental details of the P3HT: ZnO nanoparticles blend that should include the total concentration, blending time, temperature, and other specifications.
  • What was the intensity of light irradiation? Can the author please provide the light spectrum?
  • Please refer to recent studies of OPVs to improve the quality of the manuscript. 1109/JPHOTOV.2021.3074077; https://doi.org/10.1016/j.polymer.2021.123385;
  • In Figure 3, I am unable to see the clear difference in the asymmetry with and without light irradiation.
  • Is there any specific reason to choose a temperature range between 15K to 300K?

Author Response

(The authors gave the same response as above.)

Reviewer 3 Report

Please, refer to the attached document!

Author Response

(The authors gave the same response as above.)

Reviewer 4 Report

The authors have reported “Effect of Light Irradiation on the Diffusion Rate of the Charge Carrier Hopping Mechanism in P3HT: ZnO Nanoparticles Studied by μ+SR”. There is no evidence or link of the usage of this work with the solar applications as hinted by the authors in the abstract and introduction part. In addition, I found that similar kinds of works have already been reported by the same group (by referring to the twelve journals from the same group cited in the references of this draft). This work would be interesting to the researchers in this area. However, I am not convinced with the significance of this paper. For these reasons, I am unable to recommend this paper for acceptance at this stage in the Energies.

The authors can improve the manuscript before submitting it to any of the journals for publication.

  1. What is the bandgap of the ZnO and hybrid nanoparticles?
  2. As hinted by the authors in the abstract and introduction (Solar application), the authors are suggested to link the as obtained results for the solar application. How the as obtained results can impact the improvement of solar cells?
  3. The authors are suggested to provide SEM or TEM images.
  4. The authors are suggested to maintain the font in each figure identical and readable.

Author Response

(The authors gave the same response as above.)

Round 2

Reviewer 2 Report

Authors have addressed most of my comments satisfactorily. It can be proceeded for the publication after English editing.

Author Response

Authors would like to thank for valuable comments and suggestions to increase the quality of our manuscript. We made improvements for both English and quality of manuscript. We also checked of English with professional agencies for English language editing.

Reviewer 3 Report

Please, refer to the attached document!

Author Response

Point 1: The use of the English language needs to improve in some points of the text

Respond 1: We made improvements for both English and quality of manuscript. We also checked of English with professional agencies for English language editing.

Point 2: The authors do not consider the space between numbers and units. For example, in row 134 is reported 130W instead 130 W. Please put always a space.

Respond 2: We have re-checked and revised the space between numbers and units in the text.

Point 3: In “conclusion section“, the author continue not to give their opinion on the future prospective about this new model and what could be its main advantages.

Respond 3: We added the explanation about the main advantages and the future prospective at the end of conclusion section.

Point 4: Although slightly modified and justified by these authors, this article, candidate to be published on energies journal continues to have the same title, similar words, and the same simulations of the previous work. This is a fact!

Respond 4: Our previous publication is concerned with the dimensionality of charge carrier mobility in various materials without calculating the diffusion rate. The diffusion rate is very important to determine the value of charge carrier mobility which will be related to the efficiency of the solar cell. So this publication is a new publication and different from our previous publications.

Point 5: With the aim to improve the paper in the references, I noticed an oversight:

Citroni, R.; Di Paolo, F.; Livreri, P. Evaluation of an optical energy harvester for SHM application, AEU - International Journal of Electronics and Communications Volume 111, November 2019, 152918

Respond 5: We added explanation about energy harvester in introduction and cited paper of Citroni et al. (2019) as reference no. [7] to improve the quality of our manuscript.

In light of these results, this reviewer suggests minor review by taking into consideration also these comments before that this article can be published.

Respond: Authors would like to thank for valuable comments, suggestions and recommendation for minor review of our manuscript to publish in Energies.

Please see the attachment for detail.

Reviewer 4 Report

It will be publishable in the present form.

Author Response

Authors would like to thank for valuable comments and recommendation of our manuscript to publish in Energies. We made improvements for both English and quality of manuscript. We also checked of English with professional agencies for English language editing.
